# A Study on the Impact of Financial Literacy and Digital Capabilities on Entrepreneurial Intention: Mediating Effect of Entrepreneurship

**DOI:** 10.3390/bs14020121

**Published:** 2024-02-08

**Authors:** Gyung-Lan Kang, Cheol-Woo Park, Seung-Hwan Jang

**Affiliations:** 1School of Science and Technology Innovation, Pusan National University, Busan 46241, Republic of Korea; kgl1128@pusan.ac.kr; 2Department of Software, Catholic University of Pusan, Busan 46241, Republic of Korea; cwpark@cup.ac.kr

**Keywords:** financial literacy, digital capabilities, entrepreneurial intention, entrepreneurship

## Abstract

In the post-COVID-19 era, the content of work and the necessary skills are rapidly changing due to the digital transformation of the way people work. Entrepreneurial adaptability and digital capability are the most necessary competencies for exploring opportunities and quickly turning them into a professional career amid a crisis. Financial literacy is also essential for expanding skills in economic and social life. The purpose of this study is to verify the influence of university students’ financial literacy and digital capability on entrepreneurial intention and the mediating effect of entrepreneurship. To this end, a survey was conducted on university students in Busan and Gyeongnam, and a sample of 162 respondents was verified using SPSS 28.0. As a result of the study, it was found that financial literacy had a partially positive effect on entrepreneurship and entrepreneurial intention. Digital capability was found to have a positive effect on entrepreneurship and entrepreneurial intention. It was found that entrepreneurship had a partially positive effect on entrepreneurial intention. It was found that entrepreneurship had a partially positive mediating effect between financial literacy and entrepreneurial intention. It was found that entrepreneurship had a positive mediating effect between digital capability and entrepreneurial intention. As a result of this study, it was confirmed that financial literacy, digital capability, and entrepreneurship are very important competencies for university students to adapt to new trends and promote start-ups in a rapidly changing job environment after COVID-19, suggesting the need for further education.

## 1. Introduction

After the COVID-19 era, Korean society is at an important inflection point. It has succeeded in developing into an economic powerhouse, centering on large companies with technological advantages, and recently, the number of start-ups has been increasing. Innovative mold-breakers such as BTS, New Jeans, and Squid Game and trends such as meok-bang and skincare are increasing their influence on the global market. In 2021, domestic start-ups grew by 78% compared to the previous year, and the number of new jobs created by start-ups was greater than the number of jobs created by the four major companies combined [1]. However, although recent developments have been made thanks to government support systems such as the Tech Incubator Program for Start-ups (TIPS), the entrepreneurial diversity that is a driving force of creativity and resilience is still insufficient for the start-up ecosystem to spread in all directions in our society.

The WEF’s Future Jobs Report presented a mixed picture of the 2023–2027 global labor market outlook. Despite the chaos of millions of jobs being lost, the emergence of advanced technologies and increasing digital access are creating new employment opportunities with innovative products and services [2]. IT-related technologies, such as big data, cloud computing, and AI, are planned to be adopted by more than 75% of companies within the next five years and comprise 16 of the top 100 jobs on the rise, being the third-highest among all occupational groups.

In the post-COVID-19 era, not only the ways of working but also the job content, necessary skills, and replacement jobs are changing. When opportunities are taken advantage of and growth slows down, the biggest employment skill is entrepreneurial adaptability—that is, trying to find creative solutions, adapting quickly to build skills, and finding new opportunities. Developing financial understanding is also essential for expanding economic and social skills. The right knowledge, attitudes, and actions toward money increase one’s responsible decision making and willingness to act to achieve financial independence. 

Personal characteristics, environmental factors, and educational factors can influence university students’ career decisions of whether they pursue a start-up or employment. Providing experiences through financial literacy and entrepreneurship education in universities can be seen as promoting important life skills related to income activities or work life by university students entering society [3]. In addition, it is important to systematically teach start-up education because it can provide directions for career paths.

Regarding entrepreneurship, many studies have been conducted on the temporal, environmental, economic, and social backgrounds. In the unpredictable post-pandemic world, more than ever, students should be prepared to expand their life skills into self-defined jobs. In this context, research examining the dynamic relationship between digital capability and financial literacy, entrepreneurial intention, and entrepreneurship in the era of expanding digital access is of considerable significance.

The results of this study will contribute to presenting practical and policy directions necessary for students to respond to crises and adapt to new trends to have self-defined jobs, digital capability, and entrepreneurship.

## 2. Theoretical Background

### 2.1. Financial Literacy

Financial literacy is defined as the overall degree of understanding of finance, such as the financial knowledge, financial behavior, and financial attitude necessary for a reasonable and sound financial life [4]. The International Network on Financial Education (INFE), under the OECD, defines it as “a combination of cognition, knowledge, skills, attitudes, and actions necessary for individuals to make sound financial decisions and ultimately achieve their financial welfare” [5]. 

According to the Bank of Korea’s [4] “2022 National Financial Literacy Survey,” Korea’s financial literacy was 66.5 points, with 75.5 points for financial knowledge, 52.4 points for financial attitude, and 65.8 points for financial activities, indicating that activities related to long-term financial plans are weak. In particular, those in their 20s (18–29 years old) have relatively high financial knowledge at 74.9 points, but their financial attitude is 48.9 points, suggesting that early financial and economic education for teenagers should be strengthened to foster their growth as economic actors who lead healthy financial lives [5]. The comprehension components of financial literacy are shown in Table 1.

Financial literacy establishes various perspectives on economic and financial phenomena, and considering that individual economic decision making can affect the country and the world from the small social unit of family to the country, the global civic quality of implementing ethical and responsible economic actions and further solving economic problems in cooperation with the community is expanding as the main goal of financial education [6]. 

### 2.2. Digital Capabilities

Digital capabilities refers to knowledge and understanding of the nature, role, and opportunity of ICT as a required set of skills and attitudes when using digital media to perform various tasks confidently and discerningly [7]. Digital capabilities consist of a variety of skills, including using digital technology in media, communication, technology and computing, literacy, and information science, as well as evaluating digital technology critically with the right attitude to participate in digital culture [8]. 

Lee and Jeon [9] defined the digital capabilities that digital education should pursue in a modern society where digital transformation is accelerating as ‘digital knowledge, skills, and attitudes necessary to comply with responsibilities and obligations, exercise rights, and realize professionally required tasks as citizens of a digital society‘.

### 2.3. Entrepreneurship

Entrepreneurship can be defined from a wide variety of perspectives. Entrepreneurship is “discovering opportunities and embodying them to create new values,” which refers to a mindset that overcomes scarce resources and focuses on opportunities with the attitude and spirit that must be equipped in the process of pursuing profits and fulfilling social responsibilities through risk sensitivity [10]. This includes a calculated risk sensitivity attitude, efficient exploratory ability, skills to gather the necessary resources, and the ability to recognize opportunities in uncertain and chaotic situations [11]. Entrepreneurship is the ability to create innovation and new values by taking high risks, capturing opportunities, and taking initiative with respect to challenges in uncertain situations. Most scholars hold innovation, initiative, and risk sensitivity as components of entrepreneurship [12,13]. Recently, research on entrepreneurship has shifted from interest in life to implementing the combination of individuals with entrepreneurial temperament and valuable opportunities [14].

### 2.4. Entrepreneurial Intention

While some jobs will become obsolete or disappear as technological advances change the labor market, developing skills and excellent flexibility could provide aworking environment that suits various interests and preferences [15]. Entrepreneurial intention encompasses personal and background characteristics, a general level of understanding of start-ups and how the founder’s background is mobilized in start-ups. Gnyawali and Fogel [16] argued that while positive perceptions of start-ups or entrepreneurs positively affect start-up motivation, start-up efficacy, start-up will, and start-up-related activities, negative perceptions of start-ups negatively affect start-up motivation, start-up efficacy, start-up will, and related activities. Lies et al. [17] analyzed whether there was a significant statistical difference in entrepreneurial intention between two groups: those who were educated in economics and entrepreneurship and those who were not educated. As a result, it was confirmed that those who received economics and entrepreneurship education had higher entrepreneurial intention. Yoon [18] argued that a positive perception that a start-up or entrepreneur is contributing to the country and society influences the formation of a positive attitude toward start-ups, and entrepreneurship plays a positive mediating role between start-up motivation and start-up will.

### 2.5. Influence Relationship between Variables

Regarding the determinants of university students’ willingness to start a business, Jeong and Cho [19] compared the importance by classifying them into social factors, university factors, and personal factors, and argued that entrepreneurship, a personal factor, had the greatest influence on the will to start a business. Valdez-Juárez et al. [20] argued that the discovery of business opportunities and the psychological profile of individuals had an indirect effect on personal characteristics and entrepreneurial intention in the influence of university students’ personal traits and psychological profiles on entrepreneurial intention based on the theory of reasoned action and the theory of planned behavior. Kim [21] argued that as the spread of smartphones has led to the broad growth of the one-person media industry and an environment in which anyone can easily access and create media, the ability to utilize media devices has improved, digital-friendly attitudes and confidence have increased, and more diverse business opportunities can be captured. Hwang [22] empirically analyzed the effects of digital education and digital capabilities on start-up recognition, and presented research results that the higher the level of digital education and digital capabilities, the more positive effect it has on start-up recognition. Kim et al. [23] argued that the ability to utilize digital media is an essential factor in securing a competitive advantage in the start-up market. Retzmann and Seber [24] argued that individuals with economic education can consider the interests, profits, and values of others in economic interaction by making independent financial decisions and making reasonable financial choices among given alternatives and pursuing legitimate profits. Digital capability has a positive effect on the capture of new business opportunities in today’s rapidly digitalizing way of working [22,23]. Financial literacy increases the practical will to achieve the expected financial level of the future [24,25,26]. People with an entrepreneurial temperament can create new value by discovering opportunities and implementing combinations even in uncertain situations [10,14]. Putting previous studies together, it can be seen that entrepreneurship has a close correlation between digital capability, financial literacy, and entrepreneurial intention variables.

## 3. Research Design

### 3.1. Establishment of Research Model and Hypothesis

In this study, research models and hypotheses were established, as shown in Figure 1, to verify the influence of university students’ financial literacy and digital capabilities on entrepreneurial intention and the mediating effect of entrepreneurship.

Direct research on financial literacy and entrepreneurial intention is rare in Korea. Similar studies confirm that the higher a university student’s financial literacy, the more desirable financial management behavior can be [25,26]. Retzmann and Seber [24] argued that individuals with financial education can consider the interests, profits, and values of others in economic interactions by making rational financial choices and pursuing legitimate profits at the same time. Hypotheses 1 and 2 were established based on previous studies.

**Hypothesis 1 (H1).** 
*Financial literacy will have a significant positive (+) effect on entrepreneurship.*


**Hypothesis 2 (H2).** 
*Financial literacy will have a significant positive (+) effect on entrepreneurial intention.*


In the knowledge and information society, start-up investment trends account for a high proportion of investment in the Internet sector, which can be seen as idea that the higher the digital capability, the more opportunities to capture various business opportunities, which increases the entrepreneurial intention [21,22]. Kim et al. [23] argued that the ability to utilize digital media is an essential factor in securing a competitive advantage in the start-up market. Therefore, Hypotheses 3 and 4 were established.

**Hypothesis 3 (H3).** 
*Digital capabilities will have a significant positive (+) effect on entrepreneurship.*


**Hypothesis 4 (H4).** 
*Digital capabilities will have a significant positive (+) effect on entrepreneurial intention.*


Several factors, such as start-up motivation, start-up efficacy, social support, and entrepreneurship, have been studied as factors that influence the start-up intention of university students, and in most previous studies, entrepreneurship has been found to have a positive effect on the will to start a business [16,19]. Hypothesis 5 was established based on previous studies. 

**Hypothesis 5 (H5).** 
*Entrepreneurship will have a significant positive (+) effect on entrepreneurial intention.*


Entrepreneurship is the attitude and spirit that must be equipped in the process of pursuing profits and fulfilling social responsibilities through risk sensitivity ‘to discover opportunities and embody them to create new values’ [10]. People with an entrepreneurial temperament can create new value by discovering opportunities and implementing combinations even in uncertain situations. Therefore, Hypotheses 6 and 7 were established, as it was judged that entrepreneurship had a positive (+) mediating effect in the relationship between financial literacy, digital capability, and entrepreneurial intention.

**Hypothesis 6 (H6).** 
*Entrepreneurship will have a significant positive (+) mediating effect between financial literacy and entrepreneurial intention.*


**Hypothesis 7 (H7).** 
*Entrepreneurship will have a significant positive (+) mediating effect between digital capabilities and entrepreneurial intention.*


### 3.2. Construction of Measurement Tools and Data Collection

The sample selection and data collection of this study were conducted in parallel with online and offline questionnaires using Google Web for university students located in Busan and Gyeong-nam. It consisted of a total of 34 questions, and a 5-point Likert scale was used. The data collection period lasted from 1 June 2023 to 31 October 2023, and a total of 173 copies were collected, and 162 copies were used as final analysis data, excluding 11 missing copies. For the collected data, reliability and validity analysis and factor analysis were performed using the SPSS 28.0 program, and the hypothesis was verified by multiple regression analysis and mediated regression analysis. 

The questionnaire composition includes a total of 34 questions, including 14 independent variable measurement questions, 12 parameters, 1 dependent variable, and 3 demographic questions, and a 5-point Likert scale was used. The composition of the measurement questionnaire is shown in Table 2. And the operational definition of measurement variables is shown in Table 3.

## 4. Analysis of Research Results

### 4.1. Demographic Characteristics

The general characteristics of the sample are shown in Table 4. 

### 4.2. Reliability, Factor Analysis, Technical Statistics, and Correlation Analysis

In factor analysis, the factor loading value for each measurement item was analyzed from a maximum of 0.866 to a minimum of 0.608, and the eigenvalue was analyzed from a maximum of 3.3361 to a minimum of 1.220. The KMO value was above 0.7. In addition, it can be seen that the reliability was secured because Cronbach’s α coefficient for analyzing the reliability of the measurement item was above 0.7. As a result of the correlation analysis, it showed a positive (+) correlation overall, and there was a high correlation between financial knowledge, financial behavior, and digital capabilities and entrepreneurial intention. The result of the correlation analysis is shown in Table 5.

### 4.3. Test Results of Hypothesis 

#### 4.3.1. Regression Results

The results of the regression analysis are shown in Table 6.

#### 4.3.2. Mediated Regression Results

Table 7 shows the results of the Sobel test as a method of verifying the significance of the mediating effect. It was found that the mediating effect of entrepreneurship between financial literacy and entrepreneurial intention was significant in initiative and risk sensitivity, but not significant in innovation. It was verified that entrepreneurship had a significant mediating effect between digital capability and entrepreneurial intention.

### 4.4. Hypothesis Verification Results and Discussion

This study examined the relationship between financial literacy, digital capabilities, and entrepreneurship as factors influencing university students’ entrepreneurial intention. Personal characteristics and environmental and educational factors can affect university students when they are deciding beyween employment or start-ups. The development of digital technology and the convergence of heterogeneous industries are innovating the way people work and creating new jobs. In the smart digital world, financial understanding and digital capabilities help to make rational financial decisions, recognizing that start-ups create value in achieving long-term individual goals.

First, it was confirmed that financial literacy had a partially positive effect on entrepreneurship. Specifically, it was confirmed that financial knowledge had a positive effect on risk sensitivity and innovation and did not affect initiative. It was confirmed that financial behavior did not affect entrepreneurship. Financial attitudes have been confirmed to have a positive effect on entrepreneurship. Hypothesis 1 was therefore partially accepted. 

Second, it was confirmed that financial literacy had a positive effect on the entrepreneurial intention. Hypothesis 2 was therefore accepted. Financial literacy helps set long-term financial goals and work to implement them. Retzmann and Seber [24] argued that individuals with financial and economic education can make financial decisions and financial choices independently and pursue legitimate profits at the same time. Therefore, financial literacy can be interpreted as raising the entrepreneurial intention when university students decide their career paths in the present era when the job environment is changing. 

Third, Hypotheses 3 and 4 were accepted, as it was confirmed that digital competency had a positive effect on entrepreneurship and entrepreneurial intention. Jeon [28] recently argued that digital capabilities are an important factor in discovering new business opportunities and recognizing start-ups in a smart society centered on information and communication as investments in digital sectors such as mobile, digital health, and HR tech have been activated as a trend of global start-ups. 

Fourth, it was confirmed that entrepreneurship had a partially positive effect on the entrepreneurial intention. Specifically, it was confirmed that innovation had a positive effect on entrepreneurial intention, and initiative and risk sensitivity did not affect entrepreneurial intention. Hypothesis 5 was therefore partially accepted. Innovation is an attitude that actively accepts original and innovative ideas, and it can be seen that it prefers and prepares for the future rather than the present. In most previous studies, innovation has been shown to have a positive effect on entrepreneurship or entrepreneurial intention [18,27], and this study is the same. The reason that initiative and risk sensitivity did not positively affect the entrepreneurial intention is interpreted as being because the target of the sample is a university student, and the direction or perception of their future career has not yet been formed in detail. In the future, it is believed that significant results can be obtained if the sample is expanded to general adults, including graduate students.

Fifth, it was confirmed that entrepreneurship has mostly a positive mediating effect between financial literacy and entrepreneurial intention. However, Hypothesis 6 was partially accepted, as risk sensitivity was confirmed to have no mediating effect between financial knowledge and entrepreneurial intention.

Sixth, Hypothesis 7 was accepted, as entrepreneurship was found to have a positive mediating effect between digital capabilities and entrepreneurial intention. It is interpreted that entrepreneurship has a positive mediating effect in Kim [21] and Hwang [22], such that the better the ability to use digital devices, the higher an individual’s digital affinity and confidence, thus raising entrepreneurial intention through opportunity search or discovery. The results of the hypothesis test are shown in Table 8.

## 5. Conclusions

Since the COVID-19 pandemic, our society has been changing the development of new cutting-edge technologies, the way we work with digital access, and the skills necessary for it. In addition, in today’s rapidly digitizing society, digital technology is playing an increasingly important role as a facilitator of creating new ventures. This can be seen as confirming that the use of digital technology is an important success factor of entrepreneurship in several studies comparing the results of digital and non-digital entrepreneurs [29,30]. Entrepreneurial adaptability is the most necessary competency for university students, the future leaders of our society, to adapt to chaos and change, explore new opportunities, and transform into professionals. This study examined the relationship between university students’ financial literacy, digital capability, and entrepreneurial intention, focusing on the mediating effect of entrepreneurship.

The study, determined that financial literacy had a partially positive effect on entrepreneurship and entrepreneurial intention, while digital capability was found to have an overall positive effect on the same. Entrepreneurship was found to have a partially positive effect on entrepreneurial intention, including a partially positive mediating effect between financial literacy and entrepreneurial intention. It was found that entrepreneurship had an overall positive mediating effect between digital capability and entrepreneurial intention.

The academic significance and implications of this study are as follows. First, the mediating effect of entrepreneurship was examined concerning the relationship between digital capability, financial literacy, and entrepreneurial intention. Personal, background and environmental characteristics were studied as factors promoting entrepreneurship, but the relationships between digital capability and financial literacy, entrepreneurial intention, and entrepreneurship in a rapidly changing digital world were dynamically examined.

Second, it was confirmed that digital capability is an important variable in discovering new business opportunities and the intention to start a business in the information-oriented digital society. The emergence of advanced technologies and the increase in digital access are creating new employment with innovative products and services despite the confusion of millions of jobs disappearing. More than 75% of companies plan to adopt IT-related technologies such as big data, cloud computing, and AI within the next five years, making up 16 of the top 100 jobs on the rise, the third-highest number among all job groups [2]. Efforts should be made to enhance the digital affinity of digital-native university students and young people and to actively utilize them to find digital-based start-up opportunities.

Third, it suggested the need to strengthen financial literacy education in the university curriculum. Financial literacy helps individuals plan life goals in the long run and make rational decisions in various processes, and enhances responsible choices for financial independence and the practices necessary to achieve them.

As a result of this study, it was confirmed that digital capability and financial literacy are important variables in the entrepreneurial intention of university students in a rapidly changing digital transformation society. In addition, it also suggested the need to train entrepreneurship and strengthen digital capabilities and financial literacy to promote start-ups.

Despite academic and practical contributions, this study has the following limitations. First, we approached a digital transformation society that is rapidly changing due to the COVID-19 pandemic. There are several factors, such as digital literacy, digital communication, digital problem solving, and digital citizenship, but since related prior research is rare, only one upper variable for digital capability was set as a factor. In future research, more detailed and in-depth research is needed for classifying each sub-element of digital capability.

Second, this study has sample limitations. The number of university student samples in Busan and Gyeongnam was only 162, so demographic characteristics such as major and entrepreneurship experience were not included in the analysis. If the scope of the sample is expanded to general adults, including graduate students, in the future, it is expected that more meaningful research results will be produced on the effect of financial literacy and digital capability on entrepreneurial intention and the mediating effect of entrepreneurship.

As a result of this study, it will contribute to the presentation of practical and policy directions necessary for university students to adapt to new trends and have their own defined jobs in the post-COVID-19 era.

## Figures and Tables

**Figure 1 behavsci-14-00121-f001:**
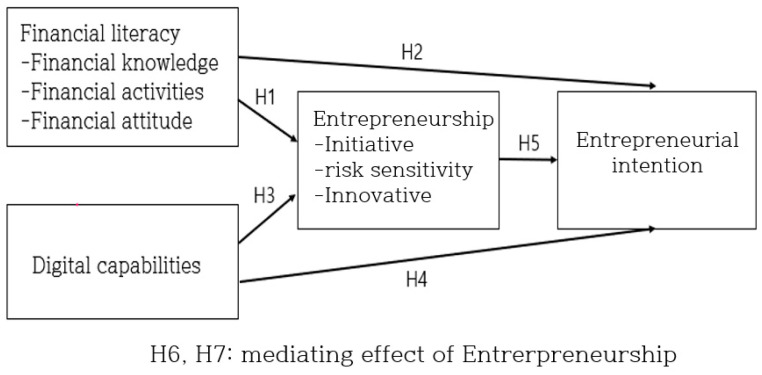
Research model.

**Table 1 behavsci-14-00121-t001:** Comprehension components of financial literacy [4].

Comprehension Components	Defining Terms	Measurement Item
Financial Knowledge	Fundamental knowledge to help compare financial instruments or services and make appropriate informed financial decisions	Inflation and purchasing power, understanding of interest concepts, simple calculation, compounding concepts, risk-to-return relationships, meaning of inflation, decentralized investment concepts
Financial Behavior	Actions performed by consumers in relation to finance, such as financial planning and budget management, and selection of financial products based on information	Household budget management efforts, active savings activities, careful purchases, timely payment of bills, usual financial inspections, long-term financial objectives, information-based financial product selection, and household balance deficit resolution
Financial Attitude	Preference for consumption and savings, present and future, the value of money’s existence, etc.	Preference for consumption over savings, preference for present over future, money exists to be spent

**Table 2 behavsci-14-00121-t002:** Composition of the measurement questionnaire.

Sortation	Analytical Factors	Number of Measurement Questions	How to Respond
Independent variable	Financial literacy	10	A five-point Likert scale
	Digital capabilities	4	
Parameters	Entrepreneurship	12	
Dependent variable	Entrepreneurial intention	5	
Demographic questions		3	
Sum		34	

**Table 3 behavsci-14-00121-t003:** Operational definition of measurement variables.

Factor	Operational Definition of Measurement Variables	Researcher
Entrepreneurship	Initiative: Plan and prepare for what needs to be done actively. The more difficult things you face, the more you try to solve them. To be considered passionate.	Yoon Nam-soo [18], Ahn Hee-Soo and Yang Dong-Woo [27]
	Risk sensitivity: Tend to push ahead with what I have to do even if it comes with risks. Perform aggressive and bold actions to achieve results. Tend to be bold when it comes to new items.	
	Innovative: Likes to challenge new tasks and tasks. Active acceptance of original and innovative ideas. Trying to find creative ideas.	
Digital capabilities	Collect the information you need through the internet. Find and utilize the information appropriate for problem solving. Effective management and utilization of the information gathered. Process financial transactions, travel reservations, taxes, etc., through the internet. Items can be purchased online.	Lee Chul-hyun and Jeon Jong-ho [9]
Financial literacy	Financial Knowledge: Inflation and purchasing power, the relationship between risk and return, and the concept of diversification investment.	OECD, INFE [5], The Bank of Korea [4]
	Financial activities: Timely payment of claims, check the financial situation, and strive to set and implement long-term financial goals. Manage personal budgets and make effective decisions.	
	Financial attitude: Preferred present over future, preferred consumption over savings, money exists to spend.	
Entrepreneurial Intention	If you start a business, you are confident that you will succeed. Want to run my own company. Running my company is the best way to improve my financial ability.	Yoon Nam-soo [18], Valdez-Juárez et al. [20]

**Table 4 behavsci-14-00121-t004:** General characteristics of the sample.

Sortation		Frequency (Number)	Percentage (%)
Gender	Man	83	64.3
	Woman	46	35.7
Start-up experience	I do	50	38.8
	I don’t	79	61.2
Major field	Social sciences	43	33.3
	Natural science	7	5.4
	Engineering field	70	31.0
	So on	39	30.2
Number of frequencies	162		

**Table 5 behavsci-14-00121-t005:** Correlation analysis.

	Financial Knowledge	Financial Behavior	Financial Attitude	Digital Capabilities	Innovation	Risk Sensitivity	Initiative	Entrepreneurial Intention
Financial knowledge	1							
Financial behavior	0.612 **	1						
Financial attitude	0.449 **	0.378 **	1					
Digital capabilities	0.634 **	0.936 **	0.460 **	1				
Innovation	0.466 **	0.321 **	0.513 **	0.388 **	1			
Risk sensitivity	0.390 **	0.260 **	0.442 **	0.320 **	0.763 **	1		
Initiative	0.313 **	0.273 **	0.424 **	0.324 **	0.682 **	0.731 **	1	
Entrepreneurial intention	0.973 **	0.638 **	0.537 **	0.665 **	0.500 **	0.405 **	0.360 **	1

** *p* < 0.01.

**Table 6 behavsci-14-00121-t006:** Regression analysis.

Sortation		Non-Standardized Coefficient		Standardized Coefficient	*t*	*sig.*	Perfect Multicollinearity Statistics	
Dependent Variable	Independent Variables	B	Standard Error	β			Tolerance Limit	VIF
Initiative	Financial knowledge	0.113	0.095	0.112	10.195	0.234	0.571	1.752
	Financial behavior	0.083	0.102	0.074	0.811	0.419	0.813	1.632
	Financial attitude	0.281	0.065	0.346	40.302	<0.001	0.781	1.280
R = 0.449 R2 = 0.202 R2adj = 0.187 F = 13.333 sig.0.001								
Risk sensitivity	Financial knowledge	0.242	0.088	0.252	20.750	0.007	0.571	1.752
	Financial behavior	−0.023	0.095	−0.021	−0.242	0.809	0.613	1.632
	Financial attitude	0.260	0.061	0.337	40.296	<0.001	0.781	1.280
R = 0.492 R2 = 0.242 R2adj = 0.227 F = 16.7800 sig.0.001								
Innovation	Financial knowledge	0.271	0.078	0.298	30.460	<0.001	0.571	1.752
	Financial behavior	−0.005	0.084	−0.005	−0.063	0.950	0.613	1.632
	Financial attitude	0.280	0.054	0.382	50.195	<0.001	0.781	1.280
R = 0.577 R2 = 0.333 R2adj = 0.320 F = 26.293 sig.0.001								
Entrepreneurial intention	Financial knowledge	0.890	0.021	0.889	420.162	<0.001	0.571	1.752
	Financial behavior	0.056	0.023	0.050	20.463	0.015	0.613	1.632
	Financial attitude	0.096	0.015	0.119	60.602	<0.001	0.781	1.280
R = 0.980 R2 = 0.960 R2adj = 0.959 F = 1262.581 sig.0.001								
Entrepreneurial intention	Initiative	0.018	0.104	0.018	0.171	0.864	0.428	2.334
	Risk sensitivity	0.049	0.124	0.046	0.390	0.697	0.335	2.987
	Innovation	0.498	0.122	0.452	40.079	<0.001	0.385	2.598
R = 0.502 R2 = 0.252 R2adj = 0.237 F = 17.704 sig.0.001								
Initiative	Digital capabilities	0.395	0.091	0.324	40.339	<0.001	1.000	1.000
R = 0.324 R2 = 0.105 R2adj = 0.100 F = 18.823 sig.0.001								
Risk sensitivity	Digital capabilities	0.369	0.087	0.320	4.266	<0.001	1.000	1.000
R = 0.320 R2 = 0.102 R2adj = 0.097 F = 18.199 sig.0.001								
Innovation	Digital capabilities	0.425	0.080	0.388	5.317	<0.001	1.000	1.000
R = 0.388 R2 = 0.150 R2adj = 0.145 F = 28.275 sig.0.001								
Entrepreneurial intention	Digital capabilities	0.803	0.071	0.665	11.248	<0.001	1.000	1.000
R = 0.665 R2 = 0.442 R2adj = 0.438 F = 126.524 sig.0.001								

**Table 7 behavsci-14-00121-t007:** Mediation effect analysis and Sobel test.

Independent/Parameter/ Dependent Variables	Mediation Effect Verification Stage	Standardized Beta Values	*t* Value	*p*-Value	R2	Sobel-Test	
						Z	*p*
Financial knowledge Initiative Entrepreneurial intention	stage 1	0.313	4.165	<0.001	0.098	2.567	0.000 ***
	stage 2	0.973	52.874	<0.001	0.946		
	stage (Independent)	0.953	50.735	<0.001	0.949		
	stage 3 (Parameter)	0.062	3.323	0.001			
Financial knowledge Risk sensitivity Entrepreneurial intention	stage 1	0.390	5.362	<0.001	0.152	1.423	0.155
	stage 2	0.973	52.874	<0.001	0.946		
	stage (Independent)	0.961	48.285	<0.001	0.947		
	stage 3 (Parameter)	0.030	1.494	0.137			
Financial knowledge Innovation Entrepreneurial intention	stage 1	0.466	6.659	<0.001	0.217	2.733	0.000 ***
	stage 2	0.973	52.874	<0.001	0.946		
	stage (Independent)	0.945	46.532	<0.001	0.949		
	stage 3 (Parameter)	0.060	2.965	0.003			
Financial behavior Initiative Entrepreneurial intention	stage 1	0.273	3.591	<0.001	0.075	2.418	0.000 ***
	stage 2	0.638	10.491	<0.001	0.408		
	stage (Independent)	0.583	9.500	<0.001	0.445		
	stage 3 (Parameter)	0.201	3.275	0.001			
Financial behavior Risk sensitivity Entrepreneurial intention	stage 1	0.260	3.407	<0.001	0.068	2.651	0.000 ***
	stage 2	0.638	10.491	<0.001	0.408		
	stage (Independent)	0.572	9.551	<0.001	0.469		
	stage 3 (Parameter)	0.256	4.277	<0.001			
Financial behavior Innovation Entrepreneurial intention	stage 1	0.321	4.288	<0.001	0.103	3.398	0.000 ***
	stage 2	0.638	10.491	<0.001	0.408		
	stage (Independent)	0.533	9.040	<0.001	0.505		
	stage 3 (Parameter)	0.329	5.585	<0.001			
Financial attitude Initiative Entrepreneurial intention	stage 1	0.424	5.926	<0.001	0.180	2.093	0.000 ***
	stage 2	0.537	8.048	<0.001	0.288		
	stage (Independent)	0.468	6.433	<0.001	0.310		
	stage 3 (Parameter)	0.162	2.225	0.027			
Financial attitude Risk sensitivity Entrepreneurial intention	stage 1	0.442	6.230	<0.001	0.195	2.603	0.000 ***
	stage 2	0.537	8.048	<0.001	0.288		
	stage (Independent)	0.445	6.115	<0.001	0.323		
	stage 3 (Parameter)	0.208	2.863	0.005			
Financial attitude Innovation Entrepreneurial intention	stage 1	0.513	7.569	<0.001	0.264	3.598	0.000 ***
	stage 2	0.537	8.048	<0.001	0.288		
	stage (Independent)	0.380	5.129	<0.001	0.357		
	stage 3 (Parameter)	0.305	4.113	<0.001			
Digital capabilities Initiative Entrepreneurial intention	stage 1	0.324	4.339	<0.001	0.105	2.255	0.000 ***
	stage 2	0.665	11.248	<0.001	0.442		
	stage (Independent)	0.612	9.980	<0.001	0.465		
	stage 3 (Parameter)	0.162	2.641	0.009			
Digital capabilities Risk sensitivity Entrepreneurial intention	stage 1	0.320	4.266	<0.001	0.102	2.725	0.000 ***
	stage 2	0.665	11.248	<0.001	0.442		
	stage (Independent)	0.596	9.903	<0.001	0.483		
	stage 3 (Parameter)	0.214	3.560	<0.001			
Digital capabilities Innovation Entrepreneurial intention	stage 1	0.388	5.317	<0.001	0.150	3.544	0.000 ***
	stage 2	0.665	11.248	<0.001	0.442		
	stage (Independent)	0.554	9.206	<0.001	0.511		
	stage 3 (Parameter)	0.286	4.745	<0.001			

*** *p* < 0.001.

**Table 8 behavsci-14-00121-t008:** Hypothesis test results.

Hypothesis		Accept Status
Hypothesis 1: Financial Literacy—Entrepreneurship		Partially Accept
1-1	Financial knowledge—Initiative	Reject
1-2	Financial knowledge—Risk sensitivity	Accept
1-3	Financial knowledge—Innovation	Accept
1-4	Financial behavior—Initiative	Reject
1-5	Financial behavior—Risk sensitivity	Reject
1-6	Financial behavior—Innovation	Reject
1-7	Financial attitude—Initiative	Accept
1-8	Financial attitude—Risk sensitivity	Accept
1-9	Financial attitude—Innovation	Accept
Hypothesis 2: Financial Literacy—Entrepreneurial Intention		Accept
2-1	Financial knowledge—Entrepreneurial intention	Accept
2-2	Financial behavior—Entrepreneurial intention	Accept
2-3	Financial attitude—Entrepreneurial intention	Accept
Hypothesis 3: Digital capabilities—Entrepreneurship		Accept
3-1	Digital capabilities—Initiative	Accept
3-2	Digital capabilities—Risk sensitivity	Accept
3-3	Digital capabilities—Innovation	Accept
Hypothesis 4: Digital capabilities—Entrepreneurial intention		Accept
Hypothesis 5: Entrepreneurship—Entrepreneurial intention		Partially Accept
5-1	Initiative—Entrepreneurial intention	Reject
5-2	Risk sensitivity—Entrepreneurial intention	Reject
5-3	Innovation—Entrepreneurial intention	Accept
Hypothesis 6: Financial Literacy—Entrepreneurship—Entrepreneurial intention (Mediating effect)		Partially Accept
6-1	Financial knowledge—Initiative—Entrepreneurial intention	Accept
6-2	Financial knowledge—Risk sensitivity—Entrepreneurial intention	Reject
6-3	Financial knowledge—Innovation—Entrepreneurial intention	Accept
6-4	Financial behavior—Initiative—Entrepreneurial intention	Accept
6-5	Financial behavior—Risk sensitivity—Entrepreneurial intention	Accept
6-6	Financial behavior—Innovation—Entrepreneurial intention	Accept
6-7	Financial attitude—Initiative—Entrepreneurial intention	Accept
6-8	Financial attitude—Risk sensitivity—Entrepreneurial intention	Accept
6-9	Financial attitude—Innovation—Entrepreneurial intention	Accept
Hypothesis 7: Digital capabilities—Entrepreneurship—Entrepreneurial intention (Mediating effect)		Accept
7-1	Digital capabilities—Initiative—Entrepreneurial intention	Accept
7-2	Digital capabilities—Risk sensitivity—Entrepreneurial intention	Accept
7-3	Digital capabilities—Innovation—Entrepreneurial intention	Accept

## Data Availability

The analyzed datasets during this study are available from the corresponding author upon reasonable request.

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
