# Peer review of "A Study on the Impact of Financial Literacy and Digital Capabilities on Entrepreneurial Intention: Mediating Effect of Entrepreneurship"

_behavsci, 2024, doi:10.3390/bs14020121_

Round 1
Reviewer 1 Report
Comments and Suggestions for Authors
Thank you for the opportunity to get acquainted with the results of your research! However, the article has significant flaws that need to be addressed before accepting for publication: 1) The wording of the topic needs to be clarified; 2) Rework of the summary to match the content of the article; 3) The references list is incomplete, poorly represented scientific publications of recent years across the world. Consequently, the analysis of the literature is weak, the theoretical basis of the proposed conceptual model is not clear; 4) The text does not include Hypothesis 5 statement; 5) The representativeness of the sample used in the study is not characterised and it is therefore difficult to assess the reliability of the results obtained.
Comments on the Quality of English LanguageModerate editing of English language required
Author Response
1) The wording of the topic needs to be clarified;
- Changed 'Entrepreneurship awareness' to 'Entrepreneurial intention'.
- 5p table 3. Entreping modified to Initiative.
2) Rework of the summary to match the content of the article;
- Delete the existing abstract and revise it as a whole as follows.
- In the post COVID-19, the content of work and the necessary skills are also rapidly changing due to the digital transformation of the way they work. Entrepreneurial adaptability and digital capability are the most necessary competencies to explore opportunities and quickly turn them into a professional career amid a crisis. Financial Literacy is also essential for expanding skills in economic and social life. The purpose of this study is to verify the influence of university students' financial literacy and digital capability on entrepreneurial intention and the mediating effect of entrepreneurship. To this end, a survey was conducted on university students in Busan and Gyeongnam, and a total of 162 respondents were verified with SPSS 28.0 as a sample. As a result of the study, it was found that financial literacy had a partially positive effect on entrepreneurship and entrepreneurship intention. Digital capability was found to have a positive effect on entrepreneurship and entrepreneurial intention. It was found that entrepreneurship had a partially positive effect on entrepreneurship intention. It was found that entrepreneurship had a partially positive mediating effect between financial literacy and entrepreneurial intention. It was found that entrepreneurship had a positive mediating effect between digital capability and entrepreneurial intention. As a result of this study, it was confirmed that financial literacy, digital capability, and entrepreneurship are very important competencies for university students to adapt to new trends and promote start-ups in a rapidly changing job environment after COVID-19, suggesting the need for education.
- Delete the existing introduction and revise it as a whole as follows.
- After COVID-19, Korean society is at an important inflection point. It has succeeded in developing into an economic powerhouse, centering on large companies with technological advantages, and recently, the number of start-ups is increasing. Innovative breakers such as BTS, New Jeans, Squid Game, Meok-bang, and Skincare are increasing their influence on the global market. In 2021, domestic start-ups grew 78% compared to the previous year, and the number of new jobs created by start-ups was greater than the number of jobs created by the four major companies combined [26]. However, recent developments have been made thanks to government support systems such as the Tech Incubator Program for Start-ups (TIPS), so the entrepreneurial diversity that is a driving force of creativity and resilience is still insufficient for the start-up ecosystem to spread in all directions in our society.
The WEF's Future Jobs Report presented a mixed picture of the 2023-2027 global labor market outlook. Despite the chaos of millions of jobs being lost, the emergence of advanced technologies and increasing digital access are creating new employment with innovative products and services [27]. IT-related technologies, such as big data, cloud computing, and AI, are planned to be adopted by more than 75% of companies within the next five years and comprise 16 of the top 100 jobs on the rise, the third-highest among all occupational groups.
In the post COVID-19, not only the way you work, but also the content of your job, the skills you need, and the jobs you replace are changing. When opportunities are taken advantage of and growth slows down, the biggest employment skill is entrepreneurial adaptability-that is, trying to find creative solutions, converting quickly to build skills, and finding new opportunities. Developing financial understanding is also essential for expanding economic and social life skills. The right knowledge, attitudes, and actions for money increase the responsible decision-making and willingness to act to achieve financial independence.
In the post COVID-19, not only the way you work, but also the content of your job, the skills you need, and the jobs that are replaced are changing. Entrepreneurial adaptability is the biggest employment skill at a time when opportunities need to be taken advantage of and rapid transition to find creative solutions, build skills by transitioning quickly, and seek new opportunities. Developing financial literacy is also essential for expanding economic and social life skills. The right knowledge, attitudes, and actions for money increase the responsible decision-making and commitment to action to achieve financial independence.
Regarding entrepreneurship, many studies have been conducted in the times, environmental, economic, and social backgrounds. In the unpredictable post-pandemic world, more than ever, students should be prepared to expand their life skills into self-defined jobs. In this context, research examining the dynamic relationship between digital capability and financial literacy, entrepreneurial intention, and entrepreneurship in the era of expanding digital access is of considerable significance.
As a result of this study, it will contribute to presenting practical and policy directions necessary for students to respond to crises and adapt to new trends to have their own defined jobs, digital capability, and entrepreneurship.
3) The references list is incomplete, poorly represented scientific publications of recent years across the world. Consequently, the analysis of the literature is weak, the theoretical basis of the proposed conceptual model is not clear;
Add recent references.
- (Saemoon Yoon, WEF Geneva, 2022.10.31.).
- (Future of Jobs Report, 2023. WEF).
4) The text does not include Hypothesis 5 statement;
Several factors such as start-up motivation, start-up efficacy, social support, and entrepreneurship have been studied as factors that influence the start-up intention of university students, and in most previous studies, entrepreneurship has been found to have a positive effect on the will to start a business. Hypothesis 5 was established based on previous studies.
Hypothesis 5. Entrepreneurship will have a significant positive (+) effect on entrepreneurial intentions.
5) The representativeness of the sample used in the study is not characterised and it is therefore difficult to assess the reliability of the results obtained.
- In order to secure additional reliability, the mediating effect Sobel-Test was performed, and the results were added below Table 7.
- Table 8 shows the results of Sobel-test as a method of verifying the significance of the mediating effect. It was found that the mediating effect of entrepreneurship between financial literacy and entrepreneurial intention was significant in initiative, risk sensitivity, and innovative was not. It was verified that entrepreneurship had a significant mediating effect between digital capability and entrepreneurial intention.
- add Table 8. Hypothesis Sobel-test results.
Reviewer 2 Report
Comments and Suggestions for Authors
The paper addresses Entrepreneurship Awareness but the measures are problematic.
1. There is no contextualization of the issue which is Entrepreneuship Awareness. Why is this a problem that needs to be addressed in the context of research?
2. There is no theory to underpin the model.
3. The review of the literature are also dated.
4. There are only 5 hypotheses in the paper but the test shows more that that which is not correct.
5. If the authors wants to test all the relationships then they should develop the hypotheses for them and dscuss the hypotheses.
6. Measurement items are problematic, there is no variable called "Entrepreneurship", it should be "Entrepreneurship Orientation", "Entrepreneurship Education",etc.
7. The "Entrepreneurship Awareness" is not measuring Awareness but is measuring Intention.
8. Data is single source data but single source data was not addressed.
9. The analysis is all incorrect as the authors tested one variable at a time.
10. Table 7 is also incorrect.
11. Need toi re-run the data using SEM and report measurement model validity and structural model
12. Need to re-look at the discussion and implicatiosn after developing the literature review.
Comments on the Quality of English LanguageSome language editing is required.
Author Response
1. There is no contextualization of the issue which is Entrepreneuship Awareness. Why is this a problem that needs to be addressed in the context of research?
- The introduction has been revised as follows.
- WEF's Future Jobs Report is ...(skip)... Money knowledge, attitudes and actions increase responsible decision-making and commitment to achieving financial independence.
2. There is no theory to underpin the model.
- As written in the paper, the hypothesis was established by referring to previous studies.
3. The review of the literature are also dated.
- Modified.
4. There are only 5 hypotheses in the paper but the test shows more that that which is not correct.
- The mediating effects of Hypotheses 6 and 7 were marked and added to the model, and the correction was completed.
5. If the authors wants to test all the relationships then they should develop the hypotheses for them and dscuss the hypotheses.
- Hypothesis was established based on previous studies.
6. Measurement items are problematic, there is no variable called "Entrepreneurship", it should be "Entrepreneurship Orientation", "Entrepreneurship Education",etc.
- Changed 'Entrepreneurship awareness' to 'Entrepreneurial intention'.
7. The "Entrepreneurship Awareness" is not measuring Awareness but is measuring Intention.
- Measure as intension in the same context as 6.
8. Data is single source data but single source data was not addressed.
- If all variables are included in the model, the correlation between independent or dependent variables is so high that the actual influence may be reduced or the causal relationship may appear as an inverse relationship, so all were separated and only relative influence was investigated with the regression model.
9. The analysis is all incorrect as the authors tested one variable at a time.
- If all variables are included in the model, the correlation between independent or dependent variables is so high that the actual influence may be reduced or the causal relationship may appear as an inverse relationship, so all were separated and only relative influence was investigated with the regression model.
10. Table 7 is also incorrect.
- Modified.
11. Need toi re-run the data using SEM and report measurement model validity and structural model
- If all variables are included in the model, the correlation between independent or dependent variables is so high that the actual influence may be reduced or the causal relationship may appear as an inverse relationship, so all were separated and only relative influence was investigated with the regression model.
12. Need to re-look at the discussion and implicatiosn after developing the literature review.
- Delete the existing conclusions and revise the conclusions as a whole as follows.
- Since the COVID-19 pandemic, our society has been changing the development of new cutting-edge technologies, the way we work with digital access, and the skills necessary for it. Entrepreneurial adaptability is the most necessary competency for university students, the future leaders, to adapt to chaos and change, explore new opportunities, and transform into a professional. This study examined the relationship between university students' financial literacy, digital capability, and entrepreneurial intention, focusing on the mediating effect of entrepreneurship.
As a result of the study, it was found that financial literacy had a partially positive effect on entrepreneurship and a positive effect on entrepreneurial intention. Digital capability was found to have a positive effect on entrepreneurship and entrepreneurial intention. Entrepreneurship was found to have a partially positive effect on entrepreneurial intention. Entrepreneurship partially showed a positive mediating effect between financial literacy and entrepreneurial intention. It was found that entrepreneurship had a positive mediating effect between digital capability and entrepreneurial intention.
The academic significance and implications of this study are as follows.
First, the mediating effect of entrepreneurship was examined on the relationship between digital capability, financial literacy, and entrepreneurial intention. Many factors such as personal characteristics, background characteristics, and environmental characteristics have been studied as factors that promote entrepreneurship, but the relationship between digital capability and financial literacy, entrepreneurial intention, and entrepreneurship in the rapidly changing digital world was dynamically examined.
Second, it was confirmed that digital capability is an important variable in discovering new business opportunities and intention to start a business in the information-oriented digital society. The emergence of advanced technologies and the increase in digital access are creating new employment with innovative products and services despite the confusion of millions of jobs disappearing. More than 75% of companies plan to adopt IT-related technologies such as big data, cloud computing, and AI within the next five years, making up 16 of the top 100 jobs on the rise, the third-highest number among all job groups [27]. Efforts should be made to enhance the digital affinity of digital-native university students and young people and to actively utilize them to find digital-based start-up opportunities.
Third, it suggested the need to strengthen financial literacy education in the university curriculum. Financial literacy helps you plan life goals in the long run and make rational decisions in various processes, and enhances responsible choices for financial independence and the practices necessary to achieve them.
As a result of this study, it was confirmed that digital capability and financial literacy are important variables in the entrepreneurial intention of university students in a rapidly changing digital transformation society. In addition, it also suggested the need to train entrepreneurship and strengthen digital capabilities and financial literacy to promote start-ups.
Despite academic and practical contributions, this study has the following limitations.
First, we approached a digital transformation society that is rapidly changing due to the COVID-19 pandemic. There are several factors such as digital literacy, digital communication, digital problem solving, and digital citizenship, but since related prior research is rare, only one upper variable for digital capability was set as a factor. In future research, more detailed and in-depth research is needed by classifying each sub-element of digital capability.
Second, this study has sample limitations. The number of university student samples in Busan and Gyeongnam was only 162, so demographic characteristics such as major and entrepreneurship experience were not included in the analysis. If the scope of the sample is expanded to general adults including graduate students in the future, it is expected that more meaningful research results will be produced on the effect of financial literacy and digital capability on entrepreneurial intention and the mediating effect of entrepreneurship.
As a result of this study, it will contribute to the presentation o practical and policy directions necessary for university students to adapt to new trends and have their own defined jobs in the post-COVID-19.
Reviewer 3 Report
Comments and Suggestions for Authors
First, I would like to congratulate the authors for the work developed. The paper is globally well-structured and the research is properly designed. The topic and objectives are convincing and interesting. Despite those aspects, the paper is not sufficiently clear regarding the lack of literature that this project is solving and what is its novelty/innovation. The literature review is generally poor and out-of-date, which can be seen from the theoretical framework and the provided support for the hypotheses and variables, which also includes, in some cases, only technical documents. This is also evident within the discussion of results. So, I would propose authors improve this aspect and enrich their research with a more in-depth analysis of the literature - also providing information for those cases for which the literature is relatively absent or missing. This can be also mentioned as the paper contribution, which is also a component of the paper that is not sufficiently highlighted.

Comments on the Quality of English LanguageThe text is generally well-written. Nonetheless, some minor improvements can be made after a more careful revision. For instance, the abstract indicates the SPSS as a sample(?) or "Hypothesis 1 was therefore 228 partially adopted", which are strange or uncommon references. These are only two examples of possible issues to be solved after a second review.
Author Response
1. What is the main question addressed by the research? The question proposed in the paper, which already underlies its title, is clearly provided to readers, which is to analyze if there is a positive mediating role in entrepreneurship concerning how financial literacy and digital capabilities affect entrepreneurship awareness in a digital Society.
- Modified as below.
- Regarding entrepreneurship, many studies have been conducted in the times, environmental, economic, and social backgrounds. In the unpredictable post-pandemic world, more than ever, universities should prepare students to expand their life skills into self-defined jobs. In this context, in the current era of expanding digital access, research examining the dynamic relationship between digital capability and financial literacy, entrepreneurial intention, and entrepreneurship can be seen as very meaningful.
As a result of this study, it will contribute to presenting practical and policy directions necessary for students to respond to crises and adapt to new trends to have their own defined jobs, digital capability, and entrepreneurship.
2. What parts do you consider original or relevant for the field? What specific gap in the field does the paper address? The mediating effect of entrepreneurship seems to be the main gap to be highlighted, and so I ask authors to strengthen the literature to reinforce this aspect in their research.
-Added content below.
- In the post COVID-19, not only the way you work, but also the content of your job, the skills you need, and the jobs that are replaced are changing. Entrepreneurial adaptability is the biggest employment skill at a time when opportunities need to be taken advantage of and rapid transition when growth slows, and new opportunities to find creative solutions and to build and grow skills through rapid transformation. Developing financial capacity is also essential for expanding economic and social life skills. Knowledge, attitudes, and actions about money increase the responsible decision-making needed for financial independence and the willingness to act to achieve it.
3. What does it add to the subject area compared with other published material? That is also an aspect that I think can be improved by adding significant /relevant literature, particularly the newest one, to signify the paper’s relevance and novelty/innovation, as I said, which is still missing.
- Added content below.
- Regarding entrepreneurship, many studies have been conducted in the times, environmental, economic, and social backgrounds. In the unpredictable post-pandemic world, more than ever, universities should prepare students to expand their life skills into self-defined jobs. In this context, in the current era of expanding digital access, research examining the dynamic relationship between digital capability and financial literacy, entrepreneurial intention, and entrepreneurship can be seen as very meaningful.
4. What specific improvements should the authors consider regarding the methodology? What further controls should be considered? Besides improving the literature, as I suggested before, if I had to recommend an improvement in this matter would be to explore the demographic variables, which is missing. I would also ask authors to further provide data by using, or justify why not is useful, SEM instead.
- Samples and research methods were described in the limitations of the study.
5. Please describe how the conclusions are or are not consistent with the evidence and arguments presented. Please also indicate if all main questions posed were addressed and by which specific experiments. Again, I think that I can only make further considerations on this with additional literature, which is not sufficiently provided by authors, globally speaking.
- Add recent references.
- (Saemoon Yoon, WEF Geneva, 2022.10.31.).
- (Future of Jobs Report, 2023. WEF).
6. Are the references appropriate? The references are overall appropriate, but relatively poor, which leads authors to justify some variables with only technical references.
Add recent references
- (Saemoon Yoon, WEF Geneva, 2022.10.31.).
- (Future of Jobs Report, 2023. WEF).
7. Please include any additional comments on the tables and figures and quality of the data. Particularly, I would suggest to change table 8 and include those findings more aligned with the research design (figure 1), which would be better from the readers’ perspective.
- Modified the contents of Table 8.
Round 2
Reviewer 1 Report
Comments and Suggestions for Authors
The adjustments made are not adequate and sufficient to accept the article for publication:
1) Changing the words "entrepreneurship awareness" to "entrepreneurial intention" in the title does not eliminate the contradictions, but creates new ones, since they are two different concepts.
2) Only the words from "entrepreneurship awareness" to "entrepreneurial intention" have been changed in the theoretical background section, but the content e.g. p.2.4 has remained unchanged. Consequently, it is incomprehensible how the authors define, understand, and measure "entrepreneurial intention";
3) If constructs are changed, the variables used to measure them should also be changed. I did not find such changes in the text, e.g. in table 2 & 3;
4) The wording of the H4 hypothesis does not correspond to the Research model in Figure 1;
5) The references are not always correct, e.g. Table 3 is a reference to Park Chul-woo and Kang Kyung-ran [24] , which explores impacts of entrepreneurship and transformational leadership on employment and startup awareness, rather than entrepreneurial intention;
6) There are practically no changes to the reference list – I would recommend looking at, e.g., the journals published by the MDPI – there have been dozens of publications on the proposed topic in recent years.
Comments on the Quality of English LanguageCould be improved.
Author Response
1) Changing the words "entrepreneurship awareness" to "entrepreneurial intention" in the title does not eliminate the contradictions, but creates new ones, since they are two different concepts.
2) Only the words from "entrepreneurship awareness" to "entrepreneurial intention" have been changed in the theoretical background section, but the content e.g. p.2.4 has remained unchanged. Consequently, it is incomprehensible how the authors define, understand, and measure "entrepreneurial intention";
-> The corrections for 1) and 2) are as follows.
-> Deleted the following sentence on page 4.
Various factors such as start-up efficacy, start-up motivation, and social support are suggested as factors that influence college students' perception of start-ups.
-> The following sentence was added on page 4 based on the latest reference.
Lies et al., [17] analyzed whether there was a significant statistical difference in entrepreneurial intention between the two groups, those who were educated in economics and entrepreneurship and those who were not educated. As a result, it was confirmed that those who received economics and entrepreneurship education had higher entrepreneurial intention.
Reference [17] Lieș, G.L.; Mureșan, I.C.; Arion, I.D.; Arion, F.H. The Influence of Economic and Entrepreneurial Education on Perception and Attitudes towards Entrepreneurship. Administrative Sciences. 2023, 13(1), 2012. https://doi.org/10.3390/admsci13100212
-> The following sentence was added on page 4 based on the latest reference.
Valdez-Juárez et al., [20] argued that the discovery of business opportunities and the psychological profile of individuals had an indirect effect on personal characteristics and entrepreneurial intention in the influence of university students' personal traits and psychological profiles on entrepreneurial intention based on the theory of reasoned action, and the theory of planned behavior.
Reference [20] Valdez-Juárez, L.E.; Ramos-Escobar, E.A.; Ruiz-Zamora, J.A.; Borboa-Álvarez, E.P. Personal and Psychological Traits of University-Going Women That Affect Opportunities and Entrepreneurial Intentions. Behavioral sciences. 2024, 14(1), 66. https://doi.org/10.3390/bs14010066
-> It is similar to the Entrepreneurial Intention questionnaire measured in the following latest paper added to the references.
Reference [20] Valdez-Juárez, L.E.; Ramos-Escobar, E.A.; Ruiz-Zamora, J.A.; Borboa-Álvarez, E.P. Personal and Psychological Traits of University-Going Women That Affect Opportunities and Entrepreneurial Intentions. Behavioral sciences. 2024, 14(1), 66. https://doi.org/10.3390/bs14010066
The main contents of the paper presented are as follows.
This construct was measured considering the theory of traits and planned behavior as complementary factors that influence the behaviors and decisions of individuals [26,37]. The construct was made up of the following 10 items: C1. I am prepared to do anything to be an entrepreneur; C2. My professional goal is to become an entrepreneur; C3. I am determined to create a company in the future; C4. I have thought very seriously about the possibility of starting a business; C5. I intend to start a company someday; C6. I intend to establish a company within 5 years of my graduation; C7. I am willing to save to invest in my own company; C8. I am interested in knowing about public financing support for entre preneurship; C9. I am willing to take advantage of business opportunities when they arise; C10. I am interested in working in a company where I can develop my entrepreneurial attitudes.
3) If constructs are changed, the variables used to measure them should also be changed. I did not find such changes in the text, e.g. in table 2 & 3;
-> Table 3 has been modified as follows.
Park Chul-woo and Kang Gung-Lan, the researcher of the entrepreneurial intention of table 3, were deleted and changed to Valdez-Juárez et al.
4) The wording of the H4 hypothesis does not correspond to the Research model in Figure 1;
-> Hypothesis 4. Entrepreneurship awareness was modified to Entrepreneurial Intention.
5) The references are not always correct, e.g. Table 3 is a reference to Park Chul-woo and Kang Kyung-ran [24] , which explores impacts of entrepreneurship and transformational leadership on employment and startup awareness, rather than entrepreneurial intention;
-> Table 3 has been modified as follows.
Park Chul-woo and Kang Gung-Lan, the researcher of the entrepreneurial intention of table 3, were deleted and changed to Valdez-Juárez et al.
6) There are practically no changes to the reference list – I would recommend looking at, e.g., the journals published by the MDPI – there have been dozens of publications on the proposed topic in recent years.
-> Added the latest two papers published in MPDI.
[17] Lieș, G.L.; Mureșan, I.C.; Arion, I.D.; Arion, F.H. The Influence of Economic and Entrepreneurial Education on Perception and Attitudes towards Entrepreneurship. Administrative Sciences. 2023, 13(1), 2012. https://doi.org/10.3390/admsci13100212
[20] Valdez-Juárez, L.E.; Ramos-Escobar, E.A.; Ruiz-Zamora, J.A.; Borboa-Álvarez, E.P. Personal and Psychological Traits of University-Going Women That Affect Opportunities and Entrepreneurial Intentions. Behavioral sciences. 2024, 14(1), 66. https://doi.org/10.3390/bs14010066
Reviewer 2 Report
Comments and Suggestions for Authors
Although the authors did not address all the concerns but the paper flows better now.
Comments on the Quality of English LanguageAcceptable.
Author Response
The paper got better because of your comments.
Thank you very much.
Reviewer 3 Report
Comments and Suggestions for Authors
Dear authors,
Thank you for the efforts and improvements made. The introduction is more complete after that, with the essential elements that should have to present the paper's relevance. Nonetheless, I think that there could be more effort to improve the literature, as suggested, since only two newest references were added. This should be used to also improve the discussion of findings. Finally, the authors could improve Table 8 in a way more aligned with my previous suggestion, which means improving the link between the results and the research design.
Comments on the Quality of English LanguageThis paragraph is repeated:
"In the post COVID-19, not only the way you work, but also the content of your job, the skills you need, and the jobs that are replaced are changing. Entrepreneurial adaptability is the biggest employment skill at a time when opportunities need to be taken advantage of and rapid transition to find creative solutions, build skills by transitioning 63 quickly, and seek new opportunities. Developing financial literacy is also essential for expanding economic and social life skills. The right knowledge, attitudes, and actions for 65 money increase the responsible decision-making and commitment to action to achieve financial independence."
Author Response
1. Thank you for the efforts and improvements made. The introduction is more complete after that, with the essential elements that should have to present the paper's relevance. Nonetheless, I think that there could be more effort to improve the literature, as suggested, since only two newest references were added. This should be used to also improve the discussion of findings.
-> -> Added the latest two papers published in MPDI.
[17] Lieș, G.L.; Mureșan, I.C.; Arion, I.D.; Arion, F.H. The Influence of Economic and Entrepreneurial Education on Perception and Attitudes towards Entrepreneurship. Administrative Sciences. 2023, 13(1), 2012. https://doi.org/10.3390/admsci13100212
[20] Valdez-Juárez, L.E.; Ramos-Escobar, E.A.; Ruiz-Zamora, J.A.; Borboa-Álvarez, E.P. Personal and Psychological Traits of University-Going Women That Affect Opportunities and Entrepreneurial Intentions. Behavioral sciences. 2024, 14(1), 66. https://doi.org/10.3390/bs14010066
2. Finally, the authors could improve Table 8 in a way more aligned with my previous suggestion, which means improving the link between the results and the research design.
-> Table 8 Sobel-test was added to the result of mediated analysis of Table 7, and the existing table 8 was deleted.
3. This paragraph is repeated:
"In the post COVID-19, not only the way you work, but also the content of your job, the skills you need, and the jobs that are replaced are changing. Entrepreneurial adaptability is the biggest employment skill at a time when opportunities need to be taken advantage of and rapid transition to find creative solutions, build skills by transitioning quickly, and seek new opportunities. Developing financial literacy is also essential for expanding economic and social life skills. The right knowledge, attitudes, and actions for money increase the responsible decision-making and commitment to action to achieve financial independence.“
-> Deleted duplicate sentences.
Round 3
Reviewer 1 Report
Comments and Suggestions for Authors
Several changes have been made to the text that improve the overall quality of the article. However, there are still gaps in the theoretical part of the study, there are few analyzed scientific articles on the topic of research - only two sources are added in the last version. Before publishing an article, it is necessary to supplement the list of sources with a Digital Object Identifier so that readers can familiarize themselves with the research to which there are references in the article.
Comments on the Quality of English LanguageOK
Author Response
Thank you very much for your review.
Two references were added to the conclusion.
Also, DOI was added to the references.
<Additional information and references>
In addition, in today's rapidly digitizing society, digital technology is playing an increasingly important role as a facilitator of creating new ventures. This can be seen as confirming that the use of digital technology is an important success factor of entrepreneurship in several studies comparing the results of digital and non-digital entrepreneurs [29-30].
[29] von Briel, F.; Davidsson, P.; Recker, J. Digital Technologies as External Enablers of New Venture Creation in the IT Hardware Sector. Entrepreneurship Theory and Practice. 2018, 42(1), 47-69. https://doi.org/10.1177/1042258717732779.
[30] Bachmann, N.; Rose, R.; Maul, V.; Hölzle, K. What makes for future entrepreneurs? The role of digital competencies for en-trepreneurial intention. Journal of Business Research. 2024, 174. 1-18. https://doi.org/10.1016/j.jbusres.2023.114481.
Reviewer 3 Report
Comments and Suggestions for Authors
Dear authors,
Thank you for the improvements made. Nonetheless, I can't see the changes I proposed regarding Table 8 (to link it with the research design - figure 1) and the discussion of the findings. Best regards.
Author Response
Thank you very much for your review.
Table 8 has been modified to match the proposed changes.
Please check the modified table 8.